# Gears in chemical reaction networks for optimizing energy transduction efficiency

**Massimo Bilancioni** ✉ **& Massimiliano Esposito** ✉

Similarly to gear systems in vehicles, most chemical reaction networks (CRNs) involved in energy transduction have at their disposal multiple transduction pathways, each characterized by distinct efficiencies. We conceptualize these pathways as 'chemical gears' and demonstrate their role in refining the second law of thermodynamics. This allows us to determine the optimal efficiency of a CRN, and the gear enabling it, solely based on its topology and operating conditions, defined by the chemical potentials of its input and output species. By suitably tuning reaction kinetics, a CRN can be engineered to self-regulate its gear settings, maintaining optimal efficiency under varying external conditions. We demonstrate this principle in a biological context with a CRN where enzymes function as gear shifters, autonomously adapting the system to achieve near-optimal efficiency across changing environments. Additionally, we analyze the gear system of an artificial molecular motor, identifying numerous counterproductive gears and providing insights into its transduction capabilities and optimization.

Significant progress has been made in analyzing open chemical reaction networks (CRNs) that transduce free energy as chemical machines. These machines work by coupling chemical reactions to generate, as a net effect, a set of effective reactions among the species in the environment (chemostats), which we name processes. CRNs enable the transfer of free energy from exergonic input processes to endergonic output processes. For this transfer to occur, some reactions must carry a flux, which places the machine out of equilibrium and leads to free energy dissipation.

Examples of simple chemical machines include enzymes[1,2], biological[3,4] and artificial[5–9] molecular motors, and supramolecular systems[10–12]. More complex chemical machines include metabolic networks, which transduce free energy to sustain life: catabolic pathways break down energetic food molecules to synthesize ATP, which is then used by anabolic pathways to perform energy-demanding cellular tasks[2,4,13–15].

In most human-made machines, such as cars or bicycles, adjustable gears are critical for efficient operation across a wide range of conditions. This raises the question: can chemical machines also have gears? A primary achievement of this paper is to show that CRNs can indeed have gears and to rigorously formalize this concept. We derive gears from elementary flux modes (EFMs)[16], a concept extensively used for analyzing metabolic pathways[17,18]. Specifically, a gear is defined as an external EFM—an EFM that consumes or produces chemostatted species.

Each gear is characterized by a transduction efficiency, which is defined as the free energy it generates in the chemostats via the output process divided by the free energy it consumes from the chemostats via the input process, see Eq. (3). Notably, this gear efficiency depends exclusively on the topology of the CRN and the chemical potentials of the chemostats. Our central finding is that these gears establish the maximum efficiency a CRN can achieve under specific operating conditions, Eq. (4), refining the limits imposed by the standard second law: The external conditions select a subset of gears available for transduction, and the gear with the highest efficiency sets the upper limit of the machine efficiency. Achieving this maximum efficiency requires adjusting the machine's gearing, which is controlled by the CRN kinetics. Ideally, the flux should be concentrated on the most efficient gear across all operating conditions, but in practice, suboptimal gears often take on some of the flux.

By refining the second law of thermodynamics using gears, our work significantly extends prior studies on the transduction efficiency

Complex Systems and Statistical Mechanics, Department of Physics and Materials Science, University of Luxembourg, 30 Avenue des Hauts-Fourneaux, L-4362 Esch-sur-Alzette, Luxembourg. ✉e-mail: massimo.bilancioni21@gmail.com; massimiliano.esposito@uni.lu

of CRNs. Previous analytical results that did not rely on flux knowledge were restricted to tightly coupled or near-equilibrium CRNs[1–3,19,20]. Our framework overcomes these limitations, enabling the study of far-from-equilibrium CRNs of any complexity.

Gears provide a benchmark for assessing the efficiency of chemical machines across operating conditions. This benchmark can be used to evaluate the degree of optimality of metabolic networks in relation to metabolic switching–a widely observed phenomenon, as documented in experimental studies[21–27], where organisms adapt their type of metabolism to changing environmental conditions. To illustrate this point, we present a biologically inspired CRN designed to optimize transduction efficiency under varying conditions by self-adjusting its gears using enzymes' regulation. We also explore the trade-off between power and efficiency.

The gear framework can also be applied to molecular motors to gain deeper insights into their transduction mechanisms. This is particularly valuable given that current synthetic molecular motors operate with very low efficiencies[8,12]. Using this approach, we characterize the gear system of the first autonomous chemically driven molecular motor[7,8]. Our analysis reveals numerous counterproductive gears, with 15 out of 20 gears hindering transduction. Furthermore, across operating conditions, we assess the motor's ability to function in both forward and reverse transduction, determine the optimal efficiency in each case, and map the reaction pathways that must be prioritized to achieve it.

## Results and discussion
### Free energy transduction in chemistry
We consider an open well-stirred CRN that includes internal species $X$, external species $Y$ that are maintained at constant concentrations in the environment (chemostatted species), and a set of reactions $\rho$. At steady state, when the concentration of the $X$ species remains constant, we assume that the net effect of the CRN on the $Y$ species can be described in terms of two processes $a$ and $b$. Chemical processes are net (stochiometrically balanced) conversions among the $Y$ species that can be realized through sequences of reactions that leave the $X$ species unchanged; see SI Sect. II for more details. For example, in Fig. 1a, the two processes sufficient to describe the CRN's activity are the unimolecular interconversions $a$: $A_+ \leftrightarrow A_-$ and $b$: $B_+ \leftrightarrow B_-$. While we consider this simple case to illustrate the theory, each $A_+$, $A_-$, $B_+$, and $B_-$ could represent a group of species rather than a single one in the general case. For example, if process $a$ corresponds to $ATP$ hydrolysis, then $A_+$ would be $ATP + H_2O$ and $A_-$ would be $ADP + P_i$. The thermodynamically spontaneous direction for the two processes is the $+ \rightarrow -$ direction, which reduces the Gibbs free energy: $\Delta_a G = \mu_{A_-} - \mu_{A_+} < 0$ for process $a$ and $\Delta_b G = \mu_{B_-} - \mu_{B_+} < 0$ for process $b$. However, if these processes are coupled, meaning they are connected to the same CRN (as in Fig. 1b), free energy transduction becomes possible, allowing a process that naturally proceeds downhill to drive another process uphill. By convention, we assume that $a$ represents the input (the driving process) and $b$ the output (the driven process): Therefore, if $\mathcal{I}_i$ denotes the steady state rate at which process $i$ occurs in the $+ \rightarrow -$ direction, we have $\mathcal{I}_a > 0$ and $\mathcal{I}_b < 0$. The resulting entropy production rate can be written as (SI Sect. II)

$$\dot{\Sigma} = \underbrace{-\mathcal{I}_a \Delta_a G}_{> 0} \; \underbrace{-\mathcal{I}_b \Delta_b G}_{< 0} > 0, \tag{1}$$

where the first (resp. second) term is the input (resp. minus the output) flux of free energy, and the associated transduction efficiency is given by:

$$0 < \eta = \frac{-\mathcal{I}_b \Delta_b G}{\mathcal{I}_a \Delta_a G} < 1. \tag{2}$$

Before proceeding, we note that the findings in this paper also hold for stochastic CRN[28] and compartmentalized systems where each compartment is well-stirred. In the former case, observables' averages are considered, whereas, in the latter, the same species in different compartments are treated as different species.

### Transduction gears
The notion of gears is derived from elementary flux modes (EFMs)[16–18]. An EFM corresponds to a sequence of reactions, each performed an integer number of times in the forward or backward direction, which satisfies the following two conditions: (1) upon completion, the sequence leaves the $X$ species unchanged and (2) upon the removal of any reaction, every sequence of the remaining reactions inevitably changes the $X$ species; see SI Sect. III for more details. Given a network, its EFMs can be identified through dedicated algorithms[29]: For linear CRNs, they are equivalent to the closed paths of the corresponding graph; for nonlinear CRNs, they generalize this notion to hypergraphs. For example, the CRN in Fig. 1b has three EFMs that intuitively correspond to its different closed paths of reactions. They are $\boldsymbol{\psi}_\alpha$, $\boldsymbol{\psi}_\beta$, and $\boldsymbol{\psi}_\gamma$, depicted in Fig. 1c and given explicitly in SI Eq. (S4). One classifies EFMs as *external* if they have the net effect of producing or consuming the $Y$ species and as internal if they do not. $\boldsymbol{\psi}_\alpha$, $\boldsymbol{\psi}_\beta$, and $\boldsymbol{\psi}_\gamma$ are external since, for example, they all consume $A_+$.

Our central claim is that external EFMs constitute the gears of a CRN performing transduction. Consequently, we refer to CRNs with a single external EFM as single-gear CRNs, and those with multiple external EFMs as multi-gear CRNs. We now proceed to show why. The transduction performed by a single gear $\boldsymbol{\psi}_\text{g}$ (external EFM) is tightly coupled: The number of times that the gear $\boldsymbol{\psi}_\text{g}$ implements processes $a$ and $b$, $m_a^\text{g}$ and $m_b^\text{g}$, is topologically determined, and the associated gear efficiency is given by:

$$\eta_\text{g} = \frac{-m_b^\text{g} \Delta_b G}{m_a^\text{g} \Delta_a G}. \tag{3}$$

For the gears $\boldsymbol{\psi}_\alpha$, $\boldsymbol{\psi}_\beta$, and $\boldsymbol{\psi}_\gamma$ in Fig. 1c, we have: $m_a^\alpha = 1$, $m_b^\alpha = -2$ and $\eta_\alpha = 2\Delta_b G / \Delta_a G$; $m_a^\beta = 1$, $m_b^\beta = -1$, and $\eta_\beta = \Delta_b G / \Delta_a G$; and $m_a^\gamma = 2$, $m_b^\gamma = 0$ and $\eta_\gamma = 0$. Gear $\boldsymbol{\psi}_\gamma$ is futile as it dissipates input but does not produce any output. We note that, depending on $m_a^\text{g}$, $m_b^\text{g}$, and the operating conditions defined by the ratio $\Delta_b G / \Delta_a G$, $\eta_\text{g}$ can be negative, bigger than one, or even infinite, when $m_a^\text{g} = 0$. We define *transducing* gears as those for which $0 < \eta_\text{g} < 1$, meaning they can individually transfer free energy from the input process to the output process. In contrast, non-transducing gears cannot perform transduction on their own. For instance, a gear with infinite efficiency ($m_a^\text{g} = 0$) engages only with the output and is thus driven in the way that dissipates output free energy. In SI VI, we demonstrate that the presence of non-transducing gears in the stationary flux always reduces the overall transduction efficiency $\eta$.

### Upper bound on the transduction efficiency
Prior to fine-tuning, the stationary flux in a transducing CRN can be a superposition of many gears and the second law in Eq. (2) provides only a broad constraint $\eta < 1$, irrespective of operating conditions, as illustrated in Fig. 1d. Our central result is that we can significantly improve this bound by leveraging the CRN's topology. In particular, by refining the second law at the level of individual gears, we derive a tighter upper limit for $\eta$, which depends on the specific operating conditions:

$$\eta \leq \max_{\eta_\text{g} < 1} \eta_\text{g}. \tag{4}$$

The right-hand side corresponds to the most efficient thermodynamically feasible gear, $\eta_\text{g} < 1$, which we term the optimal gear. The intuition behind this result is analogous to cycling: to maximize the distance

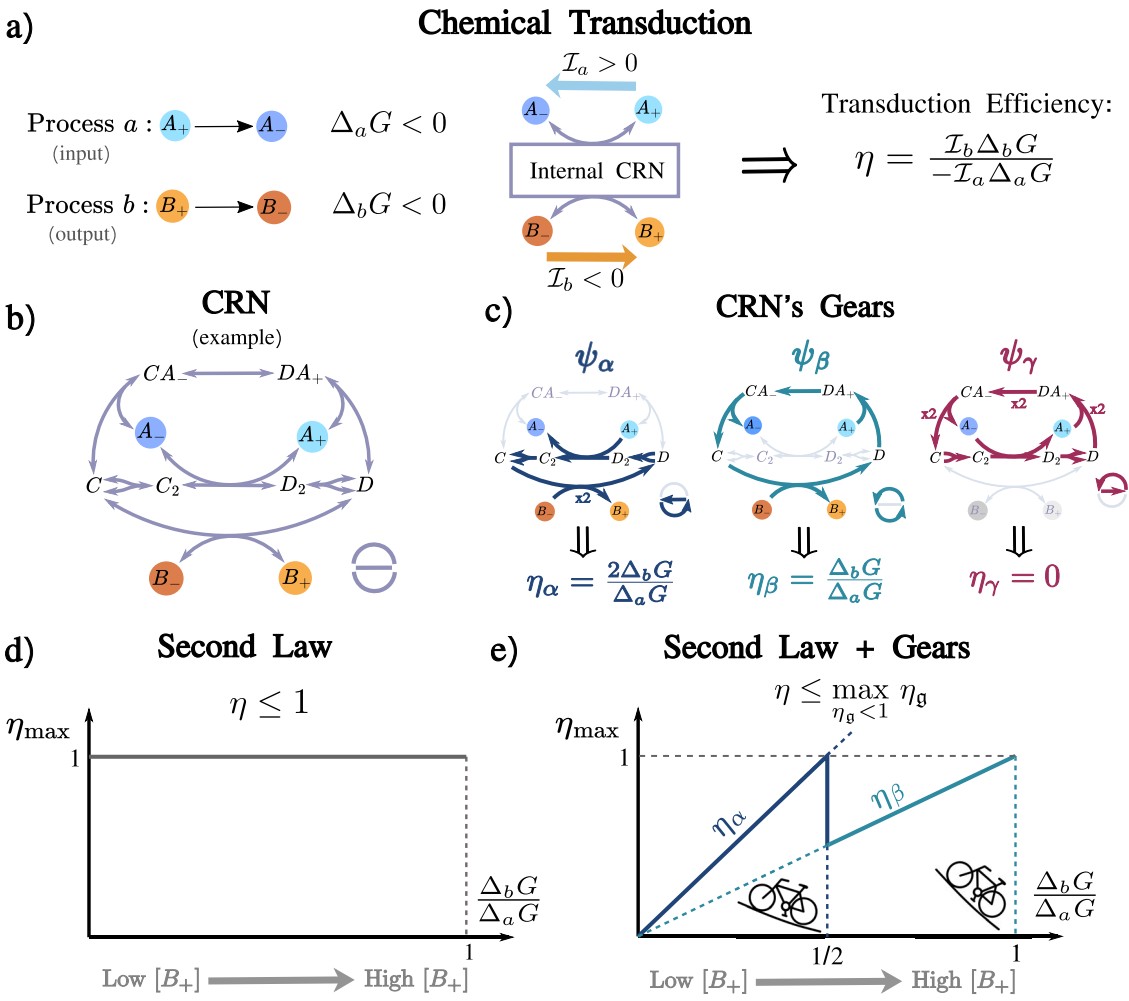

**Fig. 1 | Chemical reaction network (CRN) gears unveil the optimal transduction efficiency for any given network and under any external condition. a** Free energy transduction in a generic CRN between input process $a$ and output process $b$. The chemostatted species are colored, and the spontaneous direction of the processes is from $+\rightarrow-$, corresponding to negative Gibbs free energy changes $\Delta_a G$, $\Delta_b G < 0$. Process $a$ flows in its spontaneous direction, $\mathcal{I}_a > 0$, enabling process $b$ to flow against its spontaneous direction, $\mathcal{I}_b < 0$. The efficiency of this transduction is the ratio between the output and the input flux of free energy. **b** Example of a nonlinear multi-gear CRN. In the bottom right, a schematic representation of the CRN that highlights its structure. **c** The three gears $\psi_\alpha$, $\psi_\beta$, and $\psi_\gamma$ of the CRN (external elementary flux modes, EFMs). They correspond to the three possible closed paths of reactions and are provided explicitly in Supplementary Information

Eq. (S4). Some reactions must occur twice due to the dimerizations $C \leftrightarrow C_2$ and $D \leftrightarrow D_2$. Below, the corresponding gear's efficiencies are defined as the ratio between the # of times they produce the output $b$ and the # of times they consume the input $a$, multiplied by $\Delta_b G/\Delta_a G$. $\psi_\gamma$ is a futile gear as it only consumes the input. **d, e** Upper bounds on the transduction efficiency as a function of the load (quantified by the ratio $\Delta_b G/\Delta_a G$), imagining that the concentration of $[B_+]$ is increased. The second law for the full CRN only implies a transduction efficiency $\eta < 1$, whereas the second law refined at the gear level provides additional information: the maximum value for $\eta$ is set by the gear with the highest efficiency $\eta_g$ that is thermodynamically feasible, i.e., $\eta_g < 1$, Eq (4). At low $[B_+]$, the best gear is $\psi_\alpha$, but switches to the lighter $\psi_\beta$ as $[B_+]$ increases.

traveled per pedal stroke, one must select the heaviest gear that the terrain allows—in this context, the terrain's steepness is the analog of the ratio $\Delta_b G/\Delta_a G$. A direct implication of this inequality is that, for transduction to occur under the given operating conditions, the CRN must include at least one transducing gear $\psi_g$, i.e., $0 < \eta_g < 1$. Furthermore, equality is achieved only when all flux is concentrated on the optimal gear. In Fig. 1e, we illustrate this bound for the example CRN as a function of $\Delta_b G/\Delta_a G$. For small values of $\Delta_b G/\Delta_a G$, all gears are thermodynamically feasible, and thus the heaviest gear, $\psi_\alpha$, is the most efficient. However, under more challenging external conditions, when $\Delta_b G/\Delta_a G > 1/2$, $\psi_\alpha$ becomes thermodynamically unfeasible, and the "lighter" gear $\psi_\beta$ takes over as the most efficient.

Notably, this upper bound exhibits a discontinuity: in the regime $\Delta_b G/\Delta_a G > 1/2$, no intermediate efficiency between $\eta_\alpha$ and $\eta_\beta$ can be achieved by a superposition of the corresponding gears. This feature highlights the discrete nature of efficiency optimization in CRNs.

The proof of Eq. (4), given in SI Sect. VI, relies on the assumption that the second law is valid for each individual reaction $\rho$, i.e., the stationary flux $J_\rho$ and $-\Delta_\rho G$ have the same sign. This assumption is well known for elementary reactions, but it also encompasses any none-lementary reactions resulting from a coarse-graining of multiple elementary reactions into a single emergent cycle (see SI Sect. IA)[30]. This includes, among many other effective reactions, any enzymatic reactions occurring in the cytosol[31]. An important finding along the proof is a special decomposition of the entropy production in terms of independent gears, where all flux-force contributions are positive (SI Sect. VB).

We further note that reversing the direction of transduction, from $b$ to $a$, simply inverts the definition of gear efficiencies in Eqs. (3) and (4):

$$\eta_g^{rev} = 1/\eta_g. \tag{5}$$

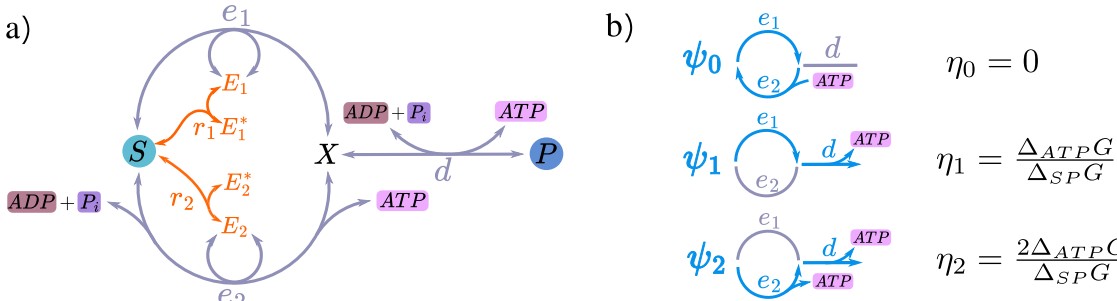

**Fig. 2 | A biologically inspired chemical reaction network (CRN) that optimizes efficiency by autonomously adjusting its gears through enzyme regulation.**
**a** The CRN is composed of two subnetworks. The gray one couples two external processes: the conversion of substrate $S$ into product $P$ and ATP hydrolysis ($H_2O$ is

omitted). The orange module is the regulatory subnetwork: $S$ inhibits the reaction $e_1$ and activates $e_2$ through the enzymes $E_1$ and $E_2$ and reactions $r_1$ and $r_2$.
**b** Schematic representation of the gears (external elementary flux modes, EFMs) $\psi_0$, $\psi_1$, and $\psi_2$ with their corresponding transduction efficiencies $\eta$; $\psi_0$ is futile.

As a result, the gear hierarchy is reversed, with the highest gear in one direction becoming the lowest in the other. Moreover, each gear remains thermodynamically viable in only one direction—either forward or backward, but not both.

### Single-gear vs. multi-gear CRNs
We now compare the multi-gear CRN depicted in Fig. 1b with the single-gear CRNs corresponding to $\psi_\alpha$ and $\psi_\beta$ using Fig. 1e. Under constant operating conditions, the optimal gear will always be, by definition, more efficient than the multi-gear CRN: $\eta_\alpha$ in the left region ($0 \leq \Delta_b G/\Delta_a G < 1/2$) and $\eta_\beta$ in the right region ($1/2 \leq \Delta_b G/\Delta_a G < 1$). However, while for a single-gear CRN operating conditions rigidly fix the direction of transduction, this direction may be controlled through kinetics in a multi-gear CRN: For example, in the range $\frac{1}{2} \leq \Delta_b G/\Delta_a G < 1$, the full CRN can either transduce from $a$ to $b$ using $\psi_\beta$, since $\eta_\beta < 1$, or from $b$ to $a$ using $\psi_\alpha$, since $\eta_\alpha^{rev} < 1$. When operating conditions vary, single-gear CRNs may exhibit suboptimal efficiency, as is the case for $\eta_\beta$ in the left region, or may only transduce within a limited range of parameters, as is the case for $\eta_\alpha$, limited to the left region. In general, lighter gears experience the first scenario, while heavier gears are prone to the second, due to their narrower operative range. In contrast, a multi-gear CRN, if kinetically well-tuned, can adapt to changes in the operating conditions. It can remain close to the optimal gear within a given region and transition to a different optimal gear in the region where the optimal gear changes. For instance, as $\Delta_b G/\Delta_a G$ exceeds 1/2, the multi-gear CRN can transition from $\eta_\alpha$ to $\eta_\beta$, maintaining efficient transduction across a wider range of conditions. Multi-geared CRN are thus more versatile. This behavior is again analogous to cycling: while a single gear might work well on a flat surface with fixed steepness, it becomes inefficient on a path with varying inclines. In such cases, the flexibility provided by multiple gears is invaluable. Similarly, combining optimality and versatility in chemistry requires a multi-gear CRN capable of autonomously switching gears in response to environmental changes.

Finally, let us note that while the analogy between CRNs and bike gears is highly useful, it is not perfect. CRNs generally operate as a weighted superposition of multiple gears. Only in a perfectly regulated CRN can gears switch discretely from one to another. In contrast, a bike typically operates with a single gear at a time. Only in poorly regulated gearing systems can the chain continuously jump between gears, creating an average effect that could be seen as a weak form of gear superposition. This distinction is not fundamental but arises from the fact that bike gearing systems are intentionally designed for energy transduction.

### Self-regulating bio-CRN
We propose a biologically inspired kinetic model of a CRN to explain how, with the help of enzymes, well-tuned gear shifting can occur. The

CRN, depicted in Fig. 2a, comprises two modules. (1) The gray subnetwork: This module couples the conversion $S \rightarrow P$ ($\Delta_{SP}G < 0$), which proceeds through an intermediate $X$, to ATP synthesis ($\Delta_{ATP}G < 0$) and includes the three gears schematically shown in Fig. 2b. Among these, only $\psi_1$ and $\psi_2$ can transduce with efficiencies $\eta_1 = \Delta_{ATP}G/\Delta_{SP}G$ and $\eta_2 = 2\Delta_{ATP}G/\Delta_{SP}G$, respectively. (2) The orange subnetwork: This module enables regulation. Specifically, $S$ inhibits the reaction $e_1$ and activates $e_2$ by inhibiting enzymes $E_1$ and promoting $E_2$. As a result, the presence of $S$ favors gear $\psi_2$ over $\psi_1$: The higher the concentration of $S$, the greater $\Delta_{SP}G$, making it more advantageous to use the heavier gear $\psi_2$. To complete the model, we assume that the forward and backward reaction fluxes (above and below the arrows) obey the mass-action law:

$$r_1 : E_1 + S \underset{k_r[E_1^*][\bar{S}]}{\overset{k_r[E_1][S]}{\rightleftharpoons}} E_1^* \qquad r_2 : E_2^* + S \underset{k_r[E_2][S]}{\overset{k_r[E_2^*][\bar{S}]}{\rightleftharpoons}} E_2,$$

$$e_1 : E_1 + S \underset{k_e[E_1][X]}{\overset{k_e[E_1][S]}{\rightleftharpoons}} X + E_1,$$

$$e_2 : E_2 + ADP + P_i + S \underset{k_e \exp(\Delta_{ATP}G)[E_2][X]}{\overset{k_e[E_2][S]}{\rightleftharpoons}} X + ATP + E_2, \qquad (6)$$

$$d : ADP + P_i + X \underset{k_d \exp(\Delta_{ATP}G)[P]}{\overset{k_d \exp(\Delta_{SP}G^0)[X]}{\rightleftharpoons}} P + ATP.$$

Here, $[\alpha]$ and $\mu_\alpha$ denote the concentration and chemical potential of species $\alpha$, with $\mu_\alpha = \mu_\alpha^0 + \log([\alpha])$ for ideal-dilute solutions (measured in units of RT). Furthermore, $\Delta_{SP}G^0 = \mu_S^0 - \mu_P^0$, $\Delta_{ATP}G = 20$ (RT), and $\mu_S^0 = \mu_X^0$. $[\bar{S}]$ is a parameter (with concentration dimension) defining the backward rates in the first two reactions. These rates satisfy the local detailed balance condition[32], ensuring thermodynamic consistency. We set the total enzymes concentrations equal, $L = [E_i^*] + [E_i]$, and solve the system assuming that reaction $d$ is much faster than reactions $e_1$ and $e_2$ (see SI Sect. VII for the calculations). In Fig. 3a, we plot the resulting efficiency $\eta$ as a function of $\Delta_{ATP}G/\Delta_{SP}G$, varying $[S]$ while keeping all the other parameters fixed. The parameter $q$ is the point on the $x$-axis where gears $\psi_1$ and $\psi_2$ are equally catalyzed (i.e., $[E_1] = [E_2]$) and can be interpreted as the gear-shifting point. It depends on $[\bar{S}]$, and to preserve transduction in the second half of the plot, we require $q < 1/2$, ensuring that gear $\psi_2$ is abandoned before it becomes unfeasible. From the three plotted values of $q$ ($q = 0.4, 0.45, 0.47$), we observe a qualitative trade-off: smaller $q$ results in lower efficiencies before $\Delta_{ATP}G/\Delta_{SP}G = 1/2$, but higher efficiencies above this value. For comparison, the black curve shows the efficiency of an unregulated CRN, obtained by removing reactions $r_1$ and $r_2$ (i.e., setting $k_r = 0$) and enforcing $[E_1] = [E_2]$ under all conditions. This unregulated CRN exhibits suboptimal efficiency in the left region and fails to transduce in the right region, where gear $\psi_2$ wastes ATP regenerating $S$. In Fig. 3b, we plot the ratio $P/P_{max}$ for the same $q$ values, where $P_{max}$ is the maximum power achievable under the constraint $[E_1]$, $[E_2] \leq L$. This

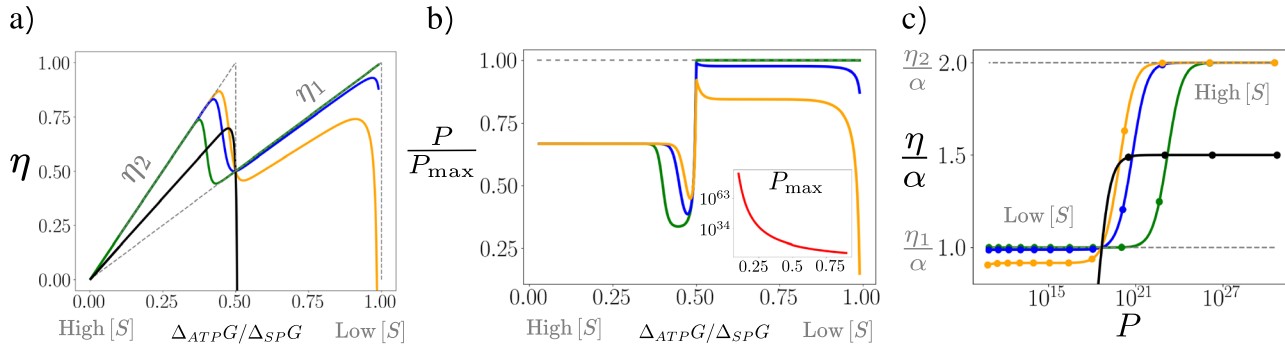

**Fig. 3 | Efficiency and power for the self-regulating chemical reaction network (CRN) in Fig. 2. a** Efficiency $\eta$ of the CRN as a function of the concentration $[S]$, represented by the ratio $\Delta_{ATP}G/\Delta_{SP}G$, with $\Delta_{ATP}G = 20$, RT, for three values of the parameter $q$ ($q = 0.4$, 0.45, and 0.47, corresponding to the green, blue, and yellow curves, respectively). $q$ is the value on the $x$-axis where $[E_1] = [E_2]$, interpreted as the effective switching point from gear $\boldsymbol{\psi}_2$ to $\boldsymbol{\psi}_1$. The black curve shows, for comparison, the efficiency of the CRN without regulation ($[E_1] = [E_2]$ for all $[S]$), while the dashed curve shows the optimal efficiency from Eq. (4). **b** The output power $P$

divided by $P_{max}$ for the previous $q$ values. $P_{max}$, shown in the inset, is the maximal power achievable under the given operating conditions with the constraint $[E_1]$, $[E_2] \leq L$. It is realized by using all the enzymes available before $\Delta_{ATP}G/\Delta_{SP}G < 1/2$ and only all the $E_1$ after. **c** Parametric plot of output power versus efficiency as a function of substrate concentration $[S]$. The different curves correspond to different $q$ values as in the other plots. The markers indicate 12 equally spaced values of $\Delta_{ATP}G/\Delta_{SP}G$ in the range [0.3, 0.9]. The vertical axis represents the efficiency rescaled by $\alpha = \Delta_{ATP}G/\Delta_{SP}G$.

normalization addresses the wide range of power values observed as a function of operating conditions due to $\Delta_{ATP}G = 20$ (RT). Smaller $q$ values result in lower output powers relative to the maximum before $\Delta_{ATP}G/\Delta_{SP}G = 1/2$, but higher output powers above that value. $P_{max}$, shown in the inset of the same figure, is a monotonically decreasing function that approaches zero as $\Delta_{ATP}G/\Delta_{SP}G \to 1$: Maximum power is achieved by fully utilizing available enzymes before $\Delta_{ATP}G/\Delta_{SP}G < 1/2$, and exclusively using $E_1$ afterwards, as gear $\boldsymbol{\psi}_2$ operates in reverse ($J_{e_2} < 0$). In Fig. 3c, we display a parametric plot of output power versus efficiency, swept by the substrate concentration $[S]$. This plot visually combines the findings of the previous two: smaller $q$ values achieve higher $P$ and $\eta$ at low $[S]$ but result in reduced $P$ and $\eta$ at high $[S]$. For comparison, we also show the unregulated CRN, which shows lower performance.

While this model focuses on substrate activation and inhibition of enzymes, it can be extended to include other types of regulation, such as product activation and inhibition by $P_i$ or ADP, ATP.

## Gears of an artificial molecular motor

We now apply our gear perspective to the first synthesized autonomous, chemically driven molecular motor, the [2]-catenane from refs. 7,8, operating clockwise and performing work against an external load. The simplified CRN describing its operation is shown in Fig. 4a (corresponding to Fig. 2 in ref. 8). It consists of six internal species ($X$), each representing a conformational state of the motor: $1_D^D$, $2_D$, $1_D^H$, $1_H^H$, $2_H$, and $1_H^D$. The fuel $F$ and waste $W$ act as chemostatted external species ($Y$). The vertical reactions correspond to mechanical motion, and the cyclic sequence $1_D^D \to 2_D \to 1_D^H \to 1_H^H \to 2_H \to 1_H^D \to 1_D^D$ describes the clockwise rotation of the motor's components (rings) relative to one another. This rotation is powered by the fuel-to-waste conversion, with $\Delta_{FW}G = \mu_W - \mu_F < 0$ (the input process), while work is extracted against an external load, with $-\Delta_{load}G > 0$ (the output process). Since the CRN forms a linear network, its EFMs correspond to all closed paths in the reaction graph. The 20 resulting EFMs, shown schematically in Fig. 4c, are all external and thus function as gears of the network. Among them, 15 are always non-transducing: $\boldsymbol{\psi}_a$, $\boldsymbol{\psi}_b$, $\boldsymbol{\psi}_c$, $\boldsymbol{\psi}_d$ are futile, as they convert fuel-to-waste without inducing directional rotation; $\boldsymbol{\psi}_e$, $\boldsymbol{\psi}_f$, $\boldsymbol{\psi}_g$, $\boldsymbol{\psi}_h$, $\boldsymbol{\psi}_i$, $\boldsymbol{\psi}_j$ exhibit a diverging gear efficiency, as they oppose the load without consuming fuel; and $\boldsymbol{\psi}_o$, $\boldsymbol{\psi}_p$, $\boldsymbol{\psi}_q$, $\boldsymbol{\psi}_r$, $\boldsymbol{\psi}_t$ dissipate both fuel and mechanical work, resulting in negative gear efficiencies. The only gears that can transduce are $\boldsymbol{\psi}_k$, $\boldsymbol{\psi}_l$, $\boldsymbol{\psi}_m$, $\boldsymbol{\psi}_n$, which involve one driving event ($F \to W$), and $\boldsymbol{\psi}_s$, which involves two ($2F \to 2W$). Their

gear efficiencies are given by:

$$\eta_k, \eta_l, \eta_m, \eta_n = \frac{\Delta_{load}G}{\Delta_{FW}G}, \qquad \eta_s = \frac{\Delta_{load}G}{2\Delta_{FW}G}. \qquad (7)$$

From Eq.(4), these gears determine the motor's maximum efficiency as a function of operating conditions given by the ratio $x = \Delta_{load}G/\Delta_{FW}G$, which is depicted by full lines in Fig. 4b. In regions where these gears are no longer thermodynamically viable in the clockwise direction, the dotted lines denote the gear efficiencies for reversed transduction, Eq. (5), when the motor rotates counterclockwise and the load drives fuel regeneration. In each regime, the gear framework pinpoints the most efficient energy conversion pathways for both forward and reverse transduction. Flux can be directed toward these pathways by appropriately tuning the molecular motor's controllable parameters—such as adjusting $F$ and $W$ concentrations, modifying chemical gating of fueling and waste-forming reactions, altering shuttling rates, changing binding site affinities, and introducing power strokes[8]. Beyond identifying these pathways, our framework also delineates the range of operating conditions where both transduction directions remain thermodynamically viable. Specifically, for $1 < x < 2$, the forward direction is feasible since $\eta_s < 1$, while the reverse is supported by $\eta_k^{rev}, \eta_l^{rev}, \eta_m^{rev}, \eta_n^{rev} < 1$. In this regime, thermodynamics does not dictate the direction of operation; rather, the direction can be controlled by adjusting the motor's kinetic parameters. Furthermore, a key insight from this analysis is that the optimal energy conversion pathways differ between forward and reverse transduction. In forward transduction ($0 < x < 1$), $\boldsymbol{\psi}_k$, $\boldsymbol{\psi}_l$, $\boldsymbol{\psi}_m$, $\boldsymbol{\psi}_n$ outperform $\boldsymbol{\psi}_s$, whereas in reverse transduction ($x > 2$), the opposite holds. As discussed below Eq. (5), this shift arises from the inversion of the gear hierarchy when transduction is reversed and holds true beyond the specific CRN considered here.

We have formalized the concept of transduction gears for CRNs, demonstrated how they can be used to identify the optimal operation of a CRN, and emphasized the role of enzymes as potential gear regulators. In this context, our work may enhance the understanding of metabolic switches, particularly regarding the extent to which they are driven by the need to maintain high thermodynamic efficiencies. Furthermore, this gear-based perspective could be instrumental in the design of artificial molecular motors by revealing their transduction possibilities and identifying the optimal gears under any regime. Looking ahead, we plan to extend this framework to encompass multi-process transduction.

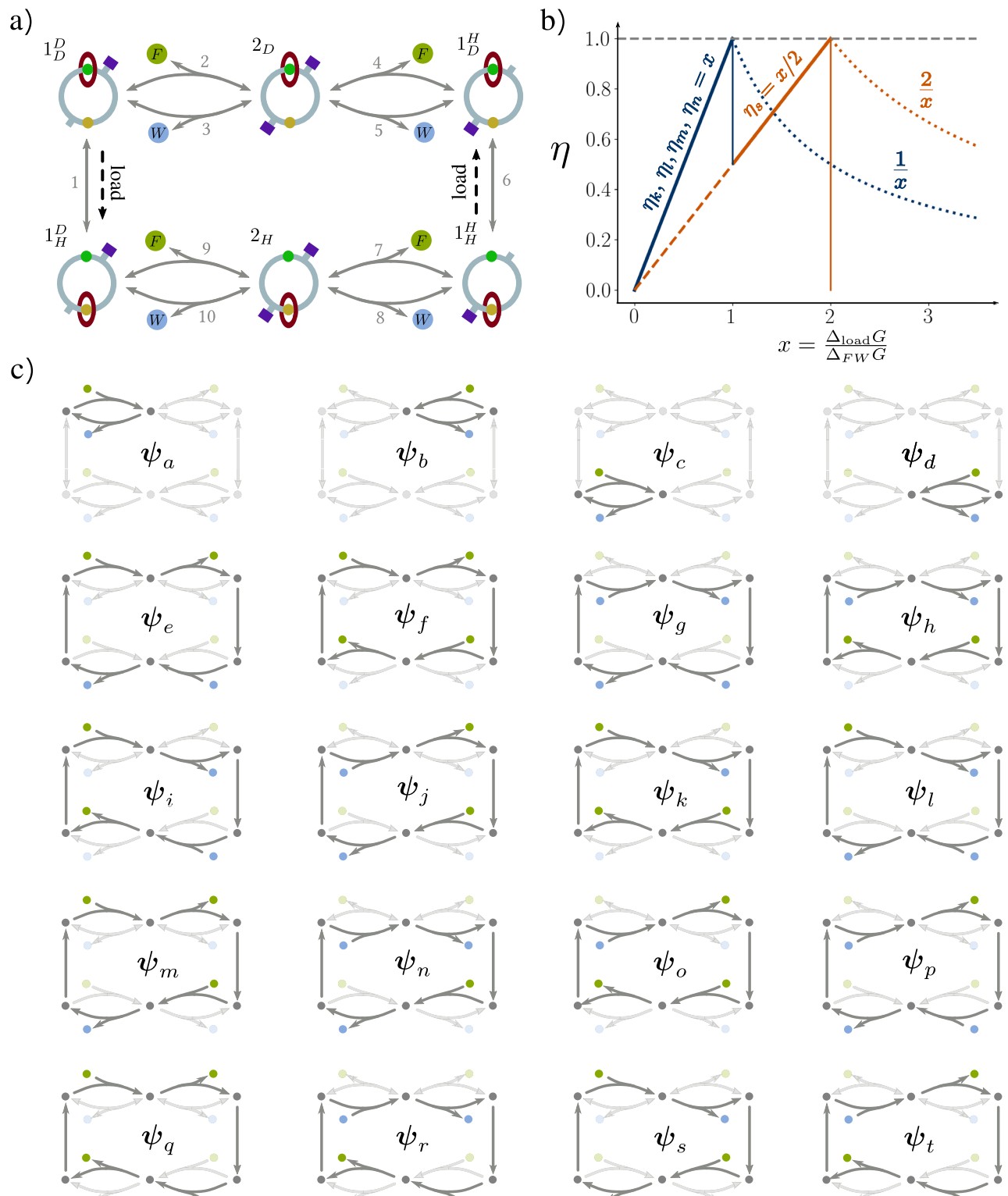

**Fig. 4 | Gears of an autonomous chemically driven molecular motor. a** Chemical reaction network (CRN) representing the motor ([2]-catenane)[8] and its six conformational states. A clockwise cyclic sequence of these states corresponds to a clockwise rotation of the motor's components (rings) relative to one another. Chemostated fuel ($F$) and waste ($W$) species drive this clockwise rotation and produce work against an external load (dashed arrows). **c** Schematic representation of the 20 gears $\psi$ (external elementary flux modes) of the CRN, each corresponding to a distinct closed path of the network. Among these, only $\boldsymbol{\psi}_k$, $\boldsymbol{\psi}_l$, $\boldsymbol{\psi}_m$, $\boldsymbol{\psi}_n$, and $\boldsymbol{\psi}_s$ can transduce fuel-to-waste conversion into work in the clockwise direction. **b** Upper bound on the transduction efficiency $\eta$ (solid line) as a function of operating conditions. The dotted lines represent the gear efficiencies in reverse transduction—when the load drives fuel regeneration—in regions where the corresponding gears are not thermodynamically viable in the forward direction.

## Data availability
All data needed to reproduce numerical results are reported in the Supplementary Information.

## Code availability
The code that generated the plots is available from the corresponding author upon request.

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

## Acknowledgements
We thank Francesco Avanzini for fruitful discussions. M.B. is funded by AFR PhD grant 15749869 funded by Luxembourg National Research Fund (FNR) and M.E. by CORE project ChemComplex (Grant No. C21/MS/16356329), and by project INTER/FNRS/20/15074473 funded by F.R.S.-FNRS (Belgium) and FNR.

## Author contributions
M.B. and M.E. contributed equally to all aspects of this work, including the conceptualization, design, analysis, and writing of the manuscript.

## Competing interests
The authors declare no competing interests.
