## [Transparent Peer Review file · Nature Communications]

Gears in Chemical Reaction Networks for Optimizing Energy Transduction Efficiency

Corresponding Author: Mr Massimo Bilancioni

Version 0:

Reviewer comments:

Reviewer #1

(Remarks to the Author)

The manuscript introduces a concept of gears in chemical reaction networks (CRNs) which provides a new language for the analyzing why some CRNs would be more capable than others of high-efficiency free energy transduction. The work is very novel, provocative, and sound. The quality of figures and writing is very high. I support publication as is.

The authors effectively draw analogies to gears on a bicycle. In both cases there is a fundamental discreteness. For the CRN gears, that discreteness emerges out of the discrete number of elementary flux modules (EFMs). Each EFM carries a well-defined efficiency, not unlike a well-defined gear ratio on a particular gear of a bicycle. Unfortunately, I think there is a significant difference between the scenarios that the analogy can obscure. The CRN steady state can involve a superposition of CRN gears. On the bicycle, an agent must select one gear at a time with the ability to toggle between them. For the CRN, an external control (like the concentration of B) can change the relative prominence of the steady state cycles around the EFMs, but it is not possible to flip from one discrete option to another. I am certain the authors clearly see the limits of the analogy. I leave it to them to decide whether it is valuable to explicitly acknowledge those limits, lest some readers over-interpret the "gears".

Reviewer #2

(Remarks to the Author)

Comments to the authors:

The manuscript "Gears in Chemical Reaction Networks: Optimizing Energy Transduction Efficiency" by Bilancioni and Esposito introduces the concept of "gears" in chemical reaction networks. They derive a general theory for such gears, which leads to a new bound on the efficiency of energy transduction in chemical reaction networks with multiple gears. They illustrate their theoretical results in a simple model for a biologically-inspired example system, and further explore a plausible mechanism for enzymatic regulation of gear-shifting. By regulating the gearing of the system, high efficiency can be achieved across a broad range of environmental conditions without significantly compromising on output power. The authors then briefly discuss applications of this framework to two models for artificial molecular motors.

This is overall a very well-written manuscript. I found it enjoyable to read, and came away understanding most of the details after my first read-through. The authors make an impressive effort to ensure their results are intuitive to the reader, though both a straightforward analogy to bicycle-riding, and a series of beautifully-designed figures which illustrate the key ideas discussed in the manuscript. The main general results, the gear decomposition and resulting bound on efficiency, are both powerful and novel; I can see many potential future directions applying this framework both to better understand existing biological CRNs, and as a design tool to improve artificial molecular machines.

The application to the bio-inspired self-regulated CRN effectively illustrates all of the main results, though as detailed below I wish the power-efficiency trade-off had been more

thoroughly explored. The applications to artificial molecular motors in the last section of the main text and the SI are intriguing, but ultimately feel incomplete in their current form. For publication in a high-profile journal such as Nature Communications, it would be nice to see a clearer demonstration of the potential of the gear framework to prove useful as a design tool for artificial molecular motors. Below I detail two substantial comments (in decreasing order of priority) which, if addressed, would significantly strengthen the manuscript. I also suggest a handful of opportunities where technical details of the manuscript could be clarified, and note a few possible typos.

Substantial Comments:

1. I was somewhat disappointed by the analysis of the artificial molecular motor in the final section of the main text. The authors consider two CRN models for artificial motors and identify the gears and their corresponding efficiencies, but do not convincingly show that the gear framework constitutes a useful lens for analyzing these systems. The model considered in the SI has a richer collection of gears than the example discussed in the main text, and might constitute a better example to lean on. Is there a simple story that can be told about how examining this motor through the lens of a geared CRN leads to a new understanding of how its performance can be improved?
2. In the section exploring the bio-CRN, the authors briefly mention the concept of a power-efficiency trade-off. However, it is not clear to me from the data shown (Figs 3a-c) whether such a trade-off occurs in this system, nor is it clear how the regulatory mechanism would influence the trade-off. To explore the power-efficiency trade-off, the authors could plot efficiency vs power parametrically as the concentration [S] is varied, showing how the self-regulated CRN performs compared to the two individual gears in isolation and/or the unregulated CRN. This could replace Fig 3c, which in my view does not provide much useful information and could be relegated to the SI.

Opportunities for Clarification:

1. The authors do not currently define the species X for the self-regulating bio-CRN example in the main text. Doing so would likely help the reader to digest this example.
2. In the brief discussion of the unregulated CRN on page 5, it might be helpful to indicate the limit required to obtain this CRN (i.e. in this case $k_r \rightarrow 0$).
3. In the "gears of an epitomic artificial molecular motor" section, the authors point out that the 'r' and 's' gears have a detrimental effect on transduction efficiency despite achieving net rotation. The authors could clarify the reasoning behind this by noting that the net rotation in these gears is driven by the load, and thus dissipates input mechanical work.
4. I found the sentence following Definition 2. in the SI to be difficult to parse. This could be made more clear by explaining the double subscript on the term $\varphi_{k\rho}$. Likewise, the concept of a "component" could be more explicitly defined.

Possible Typos:

1. Page 5, left column: "We set the total enzymes concentrations equal..."
2. Page 5, left column: "...trade-off: smaller q result in lower..."
3. Page 5, right column: "in Fig. 3 b) and c), we plot the ratio P/P_{\max} ..."
4. First sentence of SI Section III: "An elementary flux modes is a..."
5. Paragraph following Eq S19: "...of the previous two: it is $-j_c \Delta_c > 0$ for..."
6. Following Eq. S25: "... every $q_c \neq 0$ affects negatively the efficiency..."

Reviewer #3

(Remarks to the Author)

Reviewer #4

(Remarks to the Author)

This paper introduces the concept of a "chemical gear" to refine the second law of thermodynamics and offers a new perspective on the efficiency of CRNs. Through various examples, the authors effectively demonstrate that the proposed chemical gear intuitively explains why a particular chemical reaction path is favored under specific conditions and how the optimal reaction path depends on those conditions. I find this to be an interesting contribution.

The authors introduce conformational decomposition, derive its relationship to transduction efficiency and entropy production, and show that the efficiency is less than one for feasible gears and greater than zero for non-futile gears. This part represents the main contribution of the manuscript, while its interpretation and applications are discussed in the main text. If my understanding is correct, the authors define transduction efficiency, demonstrate that it is well-defined, and establish its interpretability. I find this theoretically intriguing, but can this quantity be directly measured in experiments? Otherwise, I cannot help but feel that it may have limited practical utility.

In the conclusion of the manuscript, the authors assert that "the gear-based perspective could be instrumental in the design of artificial molecular motors by addressing inefficiencies caused by futile gears and insufficient regulation." To me, this is a crucial point. However, in its current form, the manuscript does not address how to design more efficient CRNs or artificial molecular motors, which is also closely related to how transduction efficiency can be leveraged.

If this manuscript were submitted to a specialized journal, I would recommend it for publication. However, at this stage, I do not believe it meets the criteria for a high-impact journal such as Nature Communications.

Version 1:

Reviewer comments:

Reviewer #2

(Remarks to the Author)

The authors have satisfactorily addressed all of my comments, and in my opinion have likewise addressed all comments from the other reviewers. The newly added section, "gears of an artificial molecular motor", illustrates how the gear framework can be applied to study a more complex system with a dauntingly large set of gears. The theory provides a useful lens through which to understand the inefficiency of the synthetic molecular motor, and suggests principles which may improve future designs. Together with the other more minor changes, I believe this revision significantly improves on what was already a novel and well-written manuscript by showcasing the potential for broad applications. I strongly recommend publication of the manuscript in its current form.

Reviewer #3

(Remarks to the Author)

Reviewer #4

(Remarks to the Author)

All my questions are adequately addressed.

Reply to Referees' Report

Massimo Bilancioni, Massimiliano Esposito

March 15, 2025

The changes to the manuscript are highlighted in the file *Manuscript_highlighted.pdf*. Below, we report the comments of each referee (blue) and our responses (black).

Reviewer 1

The manuscript introduces a concept of gears in chemical reaction networks (CRNs) which provides a new language for the analyzing why some CRNs would be more capable than others of high-efficiency free energy transduction. The work is very novel, provocative, and sound. The quality of figures and writing is very high. I support publication as is.

We thank the reviewer for their positive assessment of our work.

The authors effectively draw analogies to gears on a bicycle. In both cases there is a fundamental discreteness. For the CRN gears, that discreteness emerges out of the discrete number of elementary flux modules (EFMs). Each EFM carries a well-defined efficiency, not unlike a well-defined gear ratio on a particular gear of a bicycle. Unfortunately, I think there is a significant difference between the scenarios that the analogy can obscure. The CRN steady state can involve a superposition of CRN gears. On the bicycle, an agent must select one gear at a time with the ability to toggle between them. For the CRN, an external control (like the concentration of B) can change the relative prominence of the steady state cycles around the EFMs, but it is not possible to flip from one discrete option to another. I am certain the authors clearly see the limits of the analogy. I leave it to them to decide whether it is valuable to explicitly acknowledge those limits, lest some readers over-interpret the "gears".

We are grateful for their insightful remarks on the analogy to bicycle gears and we agree that that analogy is not perfect. We added a short paragraph at the end of the section *Single-gear vs. Multi-gear CRNs* to clarify the limitations of this analogy. Specifically, we wrote:

“Finally, let us note that while the analogy between CRNs and bike gears is highly useful, it is not perfect. CRNs generally operate as a weighted superposition of multiple gears. Only in a perfectly regulated CRN can gears switch discretely from one to another. In contrast, a bike typically operates with a single gear at a time. Only in poorly regulated gearing systems can the chain continuously jump between gears, creating an average effect that could be seen as a weak form of gear superposition. This distinction is not

fundamental but arises from the fact that bike gearing systems are intentionally designed for energy transduction."

Reviewer 2

The manuscript "Gears in Chemical Reaction Networks: Optimizing Energy Transduction Efficiency" by Bilancioni and Esposito introduces the concept of "gears" in chemical networks. They derive a general theory for such gears, which leads to a new bound on the efficiency of energy transduction in chemical reaction networks with multiple gears. They illustrate their theoretical results in a simple model for a biologically-inspired example system, and further explore a plausible mechanism for enzymatic regulation of gear-shifting. By regulating the gearing of the system, high efficiency can be achieved across a broad range of environmental conditions without significantly compromising on output power. The authors then briefly discuss applications of this framework to two models for artificial molecular motors. This is overall a very well-written manuscript. I found it enjoyable to read, and came away understanding most of the details after my first read-through. The authors make an impressive effort to ensure their results are intuitive to the reader, though both a straightforward analogy to bicycle-riding, and a series of beautifully-designed figures which illustrate the key ideas discussed in the manuscript.

We thank the reviewer for their positive assessment of our work.

The main general results, the gear decomposition and resulting bound on efficiency, are both powerful and novel; I can see many potential future directions applying this framework both to better understand existing biological CRNs, and as a design tool to improve artificial molecular machines. The application to the bio-inspired self-regulated CRN effectively illustrates all of the main results, though as detailed below I wish the power-efficiency trade-off had been more thoroughly explored. The applications to artificial molecular motors in the last section of the main text and the SI are intriguing, but ultimately feel incomplete in their current form. For publication in a high-profile journal such as Nature Communications, it would be nice to see a clearer demonstration of the potential of the gear framework to prove useful as a design tool for artificial molecular motors. Below I detail two substantial comments (in decreasing order of priority) which, if addressed, would significantly strengthen the manuscript. I also suggest a handful of opportunities where technical details of the manuscript could be clarified, and note a few possible typos. Substantial Comments:

1. I was somewhat disappointed by the analysis of the artificial molecular motor in the final section of the main text. The authors consider two CRN models for artificial motors and identify the gears and their corresponding efficiencies, but do not convincingly show that the gear framework constitutes a useful lens for analyzing these systems. The model considered in the SI has a richer collection of gears than the example discussed in the main text, and might constitute a better example to lean on. Is there a simple story that can be told about how examining this motor through the lens of a geared CRN leads to a new understanding of how its performance can be improved?

We agree that the analysis of the example in the main text was not adding significant insight. We have now removed it and replaced it with the richer motor previously discussed in the SI. This motor is now analyzed in greater detail with an expanded discussion.

Specifically, we identify the regimes in which it can operate in forward transduction, reverse transduction, or both, and for each case, we determine the optimal energy conversion pathways (gear) and the maximum achievable efficiency. A key takeaway from this investigation is that the optimal energy conversion pathway in forward transduction differs from that in reverse transduction. This revised analysis, aimed at providing deeper insights into the transduction mechanisms of molecular motors, is presented in the final section of the paper, along with the newly added Fig. 4.

In principle, the same type of analysis performed for the self-regulating CRN in Fig. 2 could be applied to the molecular motor—introducing a kinetic model, computing efficiency as a function of operating conditions, and comparing it to the maximum efficiency in Fig. 4b. However, we chose not to pursue this here, as the resulting curve for this unregulated motor is unlikely to provide significant new insights into the experimentally studied system. Moreover, a detailed parameter optimization for a specific molecular motor falls beyond the broader theoretical scope of this work, which aims to establish a general framework for analyzing energy transduction in CRNs.

2. In the section exploring the bio-CRN, the authors briefly mention the concept of a power- efficiency trade-off. However, it is not clear to me from the data shown (Figs 3a-c) whether such a trade-off occurs in this system, nor is it clear how the regulatory mechanism would influence the trade-off. To explore the power-efficiency trade-off, the authors could plot efficiency vs power parametrically as the concentration [S] is varied, showing how the self-regulated CRN performs compared to the two individual gears in isolation and/or the unregulated CRN. This could replace Fig 3c, which in my view does not provide much useful information and could be relegated to the SI.

We have implemented this suggestion by plotting efficiency vs. power parametrically as the concentration [S] varies, including a comparison with the unregulated network. This updated analysis is now presented in Fig. 3c, and the corresponding discussion in the text has been revised accordingly.

Opportunities for Clarification:

1. The authors do not currently define the species X for the self-regulating bio-CRN example in the main text. Doing so would likely help the reader to digest this example.

We agree and now mention it in the main text.

“The grey subnetwork: This module couples the conversion $S \rightarrow P$ ($\Delta_{SP}G < 0$), which proceeds through an intermediate X , to ATP synthesis ($\Delta_{ATP}G < 0$) and ...”

2. In the brief discussion of the unregulated CRN on page 5, it might be helpful to indicate the limit required to obtain this CRN (i.e. in this case $k_r \rightarrow 0$).

We have modified the text to include this clarification.

“For comparison, the black curve shows the efficiency of an unregulated CRN, obtained by removing reactions r_1 and r_2 (i.e., setting $k_r = 0$) and enforcing $[E_1] = [E_2]$ under all conditions.”

3. In the “gears of an epitomic artificial molecular motor” section, the authors point out that the ‘r’ and ‘s’ gears have a detrimental effect on transduction efficiency despite achieving net rotation. The authors could clarify the reasoning behind this by noting that the net rotation in these gears is driven by the load, and thus dissipates input mechanical work.

We replaced that section, see the first comment. However, a similar remark is now included at the end of the *Transduction Gears* section. :

“We define *transducing* gears as those for which $0 < \eta_g < 1$, meaning they can individually transfer free energy from the input process to the output process. In contrast, *non-transducing* gears cannot perform transduction on their own. For instance, a gear with infinite efficiency ($m_a^g = 0$) engages only with the output and is thus driven in the way that dissipates output free energy. In SI VI, we demonstrate that the presence of non-transducing gears in the stationary flux always reduces the overall transduction efficiency.”

4. I found the sentence following Definition 2. in the SI to be difficult to parse. This could be made more clear by explaining the double subscript on the term $\varphi_{k\rho}$. Likewise, the concept of a “component” could be more explicitly defined.

We modified the sentence as follows:

“ One can think of it as a decomposition without cancelation: for any component ρ (i.e., a specific coordinate or entry in the vector representation), each term $\varphi_{k\rho}$ (the ρ -th entry of vector φ_k) is either zero or shares the same sign as ϕ_ρ .”

Possible Typos: 1. Page 5, left column: “We set the total enzymes concentrations equal...”

2. Page 5, left column: “... trade-off: smaller q result in lower...”

3. Page 5, right column: “in Fig. 3 b) and c), we plot the ratio P/P_{max} ...”

4. First sentence of SI Section III: “An elementary flux modes is a...”

5. Paragraph following Eq S19: “...of the previous two: it is $-j_c \Delta_c D > 0$ for...”

6. Following Eq. S25: “... every $q_c \neq 0$ affects negatively the efficiency...”

We thank the reviewer for the careful reading of our manuscript and for pointing out these typos.

Reviewer 4

This paper introduces the concept of a "chemical gear" to refine the second law of thermodynamics and offers a new perspective on the efficiency of CRNs. Through various examples, the authors effectively demonstrate that the proposed chemical gear intuitively explains why a particular chemical reaction path is favored under specific conditions and how the optimal reaction path depends on those conditions. I find this to be an interesting contribution. The authors introduce conformal decomposition, derive its relationship to transduction efficiency and entropy production, and show that the efficiency is less than one for feasible gears and greater than zero for non-futile gears. This part represents the main contribution of the manuscript, while its interpretation and applications are discussed in the main text.

If my understanding is correct, the authors define transduction efficiency, demonstrate that it is well-defined, and establish its interpretability. I find this theoretically intriguing, but can this quantity be directly measured in experiments? Otherwise, I cannot help but feel that it may have limited practical utility.

We thank the reviewer for their overall positive assessment of our work. We fully agree that experimental measurability is crucial for ensuring the broad relevance of our framework. Below, we clarify how transduction efficiency can indeed be measured in experiments.

Efficiency can be determined using two key ingredients:

(i) Chemical potential measurements: The Gibbs free energy changes of input and output species can be inferred from concentration measurements of these species, supplemented by Gibbs free energy databases, such as Equilibrator (<https://equilibrator.weizmann.ac.il/>) in biochemistry.

(ii) Flux measurements: Instead of tracking all reaction fluxes inside the network, which is often infeasible, our approach shows that it is sufficient to measure only the influx and outflux of external species: \mathcal{I}_a and \mathcal{I}_b in Eqs. (1) and (2). In a bioreactor, for instance, it suffices to track the rates at which external species enter and exit. Moreover, under stationary conditions, only a minimal set of external fluxes needs to be measured—those corresponding to the number of emergent cycles (see SI Sect. I)—as the rest can be inferred through atomic conservation. This significantly simplifies experimental validation. For example, in the CRN of Fig. 1a, despite four exchanged species, measuring just two fluxes—such as the influx of A_+ and the outflux of B_+ —is sufficient to determine \mathcal{I}_a and \mathcal{I}_b .

A concrete demonstration of this approach is already provided in [DOI: 10.1101/2024.03.21.585772] (Fig. 4d), where transduction efficiency is experimentally determined for a large variety of metabolic networks.

For molecular motors, transduction efficiency has been quantified in biological motors

such as kinesin [Svoboda et al., *Cell* 77, 773–784 (1994)] and dynein [Mallik et al., *Nature* 427, 649–652 (2004)], the latter of which functions as a geared machine. Although synthetic molecular motors typically do not operate against an external load in current experiments, partial thermodynamic efficiencies have been evaluated in synthetic systems [Amano et al., *Nature Chemistry* 14, 530–537 (2022); Corra et al., *Nature Nanotechnology* 17, 746–751 (2022)]. Given these advances, full efficiency measurements will become experimentally feasible as these molecular motors are operated against a load to perform transduction.

In the conclusion of the manuscript, the authors assert that "the gear-based perspective could be instrumental in the design of artificial molecular motors by addressing inefficiencies caused by futile gears and insufficient regulation." To me, this is a crucial point. However, in its current form, the manuscript does not address how to design more efficient CRNs or artificial molecular motors, which is also closely related to how transduction efficiency can be leveraged.

We agree that the simple molecular motor originally analyzed in the paper did not provide significant insight into designing more efficient CRNs. Reviewer 2 raised a similar concern. To address this point, we removed it and replaced it with the richer motor previously discussed in the SI, analyzing it in greater detail and expanding the discussion.

Specifically, we identify the conditions under which it can operate in forward transduction, reverse transduction, or both, and for each case, we determine the optimal energy conversion pathways (gears) and the maximum achievable efficiency. A key takeaway from this investigation is that the optimal energy conversion pathway in forward transduction differs from the one in reverse transduction. This revised analysis, aimed at providing deeper insights into the transduction mechanisms of molecular motors, is now presented in the final section of the paper alongside the newly added Fig. 4.

In principle, the same type of analysis performed for the self-regulating CRN in Fig. 2 could be applied to the molecular motor—introducing a kinetic model, computing efficiency as a function of operating conditions, and comparing it to the maximum efficiency in Fig. 4b. However, we chose not to pursue this here, as the resulting curve for this unregulated motor is unlikely to provide significant new insights into the experimentally studied system. Moreover, a detailed parameter optimization for a specific molecular motor falls beyond the broader theoretical scope of this work, which aims to establish a general framework for analyzing energy transduction in CRNs.

If this manuscript were submitted to a specialized journal, I would recommend it for publication. However, at this stage, I do not believe it meets the criteria for a high-impact journal such as *Nature Communications*.

By clarifying the experimental accessibility of efficiency measurements and demonstrating how our gear-based framework provides significantly new insight into how to optimize molecular motors depending on the mode of operation, we hope to have demonstrated the broad impact of our work.